

# Combined neural network / Phillips-Tikhonov approach to aerosol retrievals over land from the NASA Research Scanning Polarimeter

Antonio Di Noia[1], Otto P. Hasekamp[1], Lianghai Wu[1], Bastiaan van Diedenhoven[2,3], Brian Cairns[3], and John E. Yorks[4]

[1]SRON Netherlands Institute for Space Research, Sorbonnelaan 2, 3584CA Utrecht, the Netherlands
[2]Columbia University, Center for Climate Systems Research, 2910 Broadway, New York, NY 10025, United States
[3]NASA Goddard Institute for Space Studies, 2880 Broadway, New York, NY 10025, United States
[4]NASA Goddard Space Flight Center, 8800 Greenbelt Rd, Greenbelt, MD 20771, United States

*Correspondence to:* A. Di Noia (A.di.Noia@sron.nl)

**Abstract.** In this paper, an algorithm for the retrieval of aerosol and land surface properties from airborne spectropolarimetric measurements – combining neural networks and an iterative scheme based on Phillips-Tikhonov regularization – is described. The algorithm – which is an extension of a scheme previously designed for ground-based retrievals – is applied to measurements from the Research Scanning Polarimeter (RSP) onboard the NASA ER-2 aircraft. A neural network, trained on a large dataset

of synthetic measurements, is applied to perform aerosol retrievals from real RSP data, and the neural network retrievals are subsequently used as first guess for the Phillips-Tikhonov retrieval. The resulting algorithm appears capable of accurately retrieving aerosol optical thickness, fine mode effective radius and aerosol layer height from RSP data. Among the advantages of using a neural network as initial guess for an iterative algorithm are a decrease in processing time and an increase in the number of converging retrievals.

## 1 Introduction

Multiangular, multispectral measurements of intensity and linear polarization parameters of scattered Solar radiation are a useful tool for the characterization of atmospheric aerosols (Mishchenko and Travis, 1997; Hasekamp and Landgraf, 2007). The recognition of this has led to the development of a number of remote sensing instruments with spectropolarimetric capability (Kokhanovsky et al., 2015). The Polarization and Directionality of the Earth's Reflectance (POLDER)-1,2 and 3 satellite

instruments (Deschamps et al., 1994), mounted onboard the Japanese satellite Advanced Earth Observing Satellite (ADEOS) and on the French satellite Polarization & Anisotropy of Reflectances for Atmospheric Sciences coupled with Observations from a Lidar (PARASOL), have so far been the only instruments to perform in-orbit multiangle spectropolarimetric measurements. Decommissioned in 2013, POLDER-3 will be followed by the Multi-viewing Multi-channel Multi-polarization Imaging (3MI) instrument (Marbach et al., 2013), expected for launch onboard the EUMETSAT MetOp-SG satellite in 2021. While

no satellite multiangle spectropolarimeters are currently operating, research is being carried out in the development of innovative spectropolarimetric instruments, which are currently operated on aircrafts, in view of future satellite missions. These include the Research Scanning Polarimeter (RSP, Cairns et al., 1999), the Airborne Multi-angle SpectroPolarimetric Imager



(AirMSPI, Diner et al., 2013), and the Dutch instrument SPEX (van Amerongen et al., 2016). Additional spectropolarimetric missions under development are the CubeSat HyperAngular Rainbow Polarimeter (HARP) mission (Martins et al., 2014) and the Ukrainian project Aerosol-UA (Milinevsky et al., 2016).

The aerosol properties are retrieved from multiangle spectropolarimetric measurements by comparing such measurements
to forward model simulations. Retrieval methods developed so far include lookup-tables (LUTs, Deuzé et al., 2000, 2001) and iterative methods based on accurate forward modeling (Waquet et al., 2009; Dubovik et al., 2011; Hasekamp et al., 2011; Knobelspiesse et al., 2011; Wu et al., 2015). In most of these methods a LUT is used to generate a first guess for the retrieved quantities, and an iterative algorithm (e.g. maximum a posteriori, Phillips-Tikhonov regularization) is then used in order to generate solutions that fit the measurements better than the first guess.

In an experiment performed on ground-based spectropolarimetric measurements, it has been shown (Di Noia et al., 2015) that the final result of an iterative aerosol retrieval may depend on the choice of the first guess, and that replacing a LUT-based first guess with a neural network algorithm may be beneficial for the algorithm convergence and computation time, as it is relatively simple to design neural networks that provide quicker and more accurate first guess retrievals than reasonably sized LUTs with a modest computational effort. Extending this approach to aircraft and satellite measurements is possible, in principle,
once a method is devised for taking the variability of the observation geometry into account when a training set for the neural network is generated. This problem is less important – possibly absent – when working with ground-based observations. In such case, in fact, care may be taken so that a spectropolarimeter is operated only in the principal plane and at a predefined set of viewing angles. In this way, the solar zenith angle is the only variable determining the measurement geometry, and this can be easily taken into account in the neural network design process. When dealing with an airborne or satellite instrument, it is
not possible to assume that the observation geometry will always be the same. This is especially true for instruments that have a swath (e.g., POLDER, AirMSPI), in which each pixel is observed at a different set of viewing zenith and azimuth angles, whereas instruments that do not possess a swath, such as RSP, represent a situation of intermediate complexity. While it is safe to assume that an RSP measurement is always made at the same set of viewing zenith angles, the azimuth angles at which the scene is observed will be variable, and this needs to be taken into account when designing a neural network algorithm for
processing this type of measurements.

In this paper we show an application of neural network retrievals to RSP measurements. We trained a neural network on a large set of simulated RSP measurements generated for a large number of combinations of aerosol and surface parameters and observation geometries. We then used the neural network retrievals as first guess for an iterative algorithm based on the Phillips-Tikhonov method, described in Wu et al. (2015), to perform aerosol retrievals from RSP measurements acquired during
the Polarimeter Definition Experiment (PODEX) and Studies of Emission and Atmospheric Composition, Clouds and Climate Coupling by Regional Surveys (SEAC$^4$RS) measurement campaigns.

This paper is structured as follows. In Section 2 a brief description of the RSP instrument is given. In Section 3 some theoretical background is given on the use of neural networks as parameter estimation methods. Section 4 describes the design of the neural network algorithm for RSP and its validation on synthetic data. In Section 5 the results of the application of the
Phillips-Tikhonov algorithm with neural-network-based initialization to RSP measurements are discussed. In particular, the



results of comparisons between RSP retrievals and correlative data from the Aerosol Robotic Network (AERONET, Holben et al., 1998) and from the Cloud Physics Lidar (CPL, McGill et al., 2002) are presented. Finally, in Section 6 conclusions are drawn.

## 2   The NASA Research Scanning Polarimeter

The Research Scanning Polarimeter (RSP) is an airborne multi-angle spectropolarimeter initially designed as a prototype for the Aerosol Polarimeter Sensor (APS) to be launched on the Glory satellite mission (Mishchenko et al., 2007) in 2011, aimed at observing aerosols and clouds with an unprecedented accuracy by measuring intensity and linear polarization parameters of backscattered sunlight at a very high number of viewing angles (approximately 250) and in nine spectral bands (from 410 nm to 2250 nm). While the APS instrument has ultimately not reached its orbit as a consequence of the failure of the Glory

satellite launch, the RSP instrument is still being operated onboard the NASA ER-2 high altitude aircraft. RSP performs its measurements at 152 viewing angles, at the same nine spectral bands as APS (410, 470, 550, 670, 865, 960, 1590, 1880 and 2250 nm). The bands from 410 to 875 nm are particularly sensitive to aerosols. The 960 nm and 1880 nm channels are used to characterize water vapour and cirrus clouds respectively, whereas the 1590 and 2250 nm channels can be used to characterize land surface and coarse mode aerosols (Wu et al., 2015). RSP can cover an angular range going from $-60°$ to $60°$ with respect

to the aircraft vertical, where the minus sign indicates viewing directions pointed aftward with respect to the aircraft. However, because of the position of the instrument with respect to the aircraft, part the forward viewing directions from $40°$ to $60°$ are blocked by the aircraft body. The ground pixel size of an RSP measurement at nadir is 277 m. The radiometric uncertainty of RSP measurements is approximately 2% and its polarimetric uncertainty is around 0.5% (Cairns et al., 1999).

## 3   Neural network retrievals and their relationship to conventional retrievals

The retrieval of a vector of geophysical parameters $\boldsymbol{x}$ (state vector) from a vector of measurements $\boldsymbol{y}$ relies on the assumption that the measurements are related to the state vector by means of a forward model $\boldsymbol{y} = \boldsymbol{F}(\boldsymbol{x}, \boldsymbol{b}) + \boldsymbol{\epsilon}$, where $\boldsymbol{F}$ is a vector-valued deterministic function, $\boldsymbol{b}$ is a vector of parameters of $\boldsymbol{F}$ that are not included in $\boldsymbol{x}$ (e.g., observation angles, non retrieved meteorological variables, etc.), and $\boldsymbol{\epsilon}$ is the measurement noise, which is often treated as a multivariate Gaussian variable with zero mean and covariance matrix $\boldsymbol{S}_{\epsilon}$.

The conventional way of retrieving $\boldsymbol{x}$ from $\boldsymbol{y}$ consists in finding the value of $\boldsymbol{x}$ that yields the best agreement between $\boldsymbol{y}$ and $\boldsymbol{F}(\boldsymbol{x}, \boldsymbol{b})$. To this aim, a cost function depending on the discrepancy between $\boldsymbol{y}$ and $\boldsymbol{F}(\boldsymbol{x}, \boldsymbol{b})$ is defined and minimized with respect to $\boldsymbol{x}$. Because of the presence of measurement noise, the cost function is usually minimized in the least square sense. Furthermore, given the nonuniqueness of the solution due to the fact that $\boldsymbol{F}$ is – in most cases – not a biunivocal relationship, the cost function may contain some regularization terms chosen in such a way to penalize solutions that are unphysical or

unlikely (Rodgers, 2000). In general, the solution to a retrieval problem can be chosen as

$$\hat{\boldsymbol{x}} = \arg\min_{\boldsymbol{x}} J(\boldsymbol{x}, \boldsymbol{y}, \boldsymbol{b}, \boldsymbol{c}) \tag{1}$$



where $J$ is the chosen cost function and $c$ is a vector of parameters of the cost function (e.g., regularization parameters, regularization matrix, *a priori* error, etc.).

An alternative approach to the retrieval is based on the fact that, because of noise and nonuniqueness, an inverse relationship between $x$ and $y$ can be defined only in a statistical sense, and the conditional expectation $E[x|y, b]$ can be used as an estimator for $x$. Such conditional expectation is a function of $y$ and $b$, but its functional form is not known in advance. However, under the reasonable assumption that such a function is smooth, it can be estimated by collecting a number of coincidences between $x$, $y$ and $b$ (e.g., by multiple runs of a forward model), and using them to fit some smooth function $g(y, b, w)$, where $w$ is a set of parameters of the chosen function $g$ that must be determined during the fitting phase. If it reasonable to assume that the sought relationship is linear, the problem simply reduces to a linear regression of $x$ on $y$ and $b$, an approach that is quite common – for instance – in atmospheric profile retrievals from infrared and microwave measurements (Smith et al., 1970; Jackson et al., 2006). If, instead, a linear relationship between $E[x|y, b]$ and the other quantities cannot be assumed, a more general form for $g$ must be chosen. For this purpose, neural network models are usually a good choice, for at least two reasons: (i) neural network functions with at least a nonlinear hidden layer can approximate any continuous function on a compact set to an arbitrary accuracy (Hornik et al., 1989; Leshno et al., 1993); (ii) as the dimension of the training set tends to infinity, a neural network trained with the sum of squares error cost function tends to approximate the conditional expectation of the target quantity given the input vector, provided that the training samples are independent and identically distributed (Bishop, 1995a).

The latter concept can be used as a guiding principle when deciding which quantities should be used as inputs to a neural network method aimed at retrieving a certain set of geophysical parameters. These quantities do not only include the measurement vector $y$, but they also include the known parameters $b$ affecting the value of $y$. For instance, in the case of a spectropolarimetric retrieval, $y$ is a vector of reflectances and degrees of linear polarization (or eventually polarized reflectances, or Stokes parameters) measured at a number of angles and wavelengths. The value of each measurement is not only affected by the geophysical parameters we are trying to retrieve, but is also affected by quantities such as the solar zenith angle, the viewing zenith angle and the relative azimuth angle (or, equivalently, the scattering angle). When the goal is to design a neural network capable of working for several combinations of these parameters, they should in principle be used as input quantities for the network. Formally, in this way the training set will be a sample drawn from the joint probability distribution of $y$ and $b$, and the neural network retrieval will actually be a nonlinear regression of the state vector $x$ on $y$ and $b$.

Several examples of application of neural networks in parameter retrieval algorithms from remotely sensed measurements exist (Krasnopolsky, 2007). The advantages of neural networks over traditional linear regression in statistical retrievals are discussed in Del Frate and Schiavon (1998a, b). Applications to aerosol retrievals are presented, for instance, in Radosavljevic et al. (2010), Taylor et al. (2014), and Chimot et al. (2017).





## 4 Neural network retrieval scheme for RSP: Design and performance on synthetic data

The procedure followed in the neural network retrieval scheme builds further on the work described in Di Noia et al. (2015) for groundSPEX measurements. The main differences are that surface properties and additional aerosol parameters (fine and coarse mode effective variance, fraction of spherical particles for coarse mode, aerosol layer height) have been included in the retrieval state vector, and the variability of RSP observing conditions has been taken into account in the input vector.

As mentioned in Section 2, the viewing zenith angle (VZA) range of RSP measurements goes from $60°$ aftward to $40°$ forward. This asymmetry is due to the blockage of part of the forward viewing directions by the ER-2 aircraft structure. In order to simplify the generation of a training set for the neural network scheme, we decided to use a symmetrical VZA range, going from $40°$ aftward to $40°$ forward. The remaining part of the VZA range is used, instead, in the iterative retrieval that uses the neural network output as first guess, as explained in Section 5. As for the relative azimuth angles of RSP measurements we have assumed that, for each multiangular observation, the measurements at forward-viewing angles are made with a common relative azimuth angle $\varphi$ and the corresponding aftward-viewing measurements are made at an angle $\varphi + 180°$, with $\varphi$ depending on the particular flight leg. We observed from real data that this is a good approximation for most RSP measurements. As a result of this assumption, the scalar value $\varphi$ is the only variable we used in order to characterize the RSP viewing geometry in the neural network input vector. In order to avoid the issue of characterizing water-vapour absorption in the training set, we only used the first five RSP channels (410, 470, 550, 670, 865 nm). We assumed the "Ross Thick - Li Sparse reciprocal combination", hereinafter referred to as "Ross-Li model" (Maignan et al., 2004, and references therein), for the surface bidirectional reflectance distribution function (BRDF) and the model by Maignan et al. (2009) for the bidirectional polarization distribution function (BPDF).

In the Ross-Li model, the surface BRDF $R$ is expressed by means of the following linear combination (Lucht et al., 2000):

$$R(\theta_s, \theta_v, \varphi, \lambda) = f_{\text{iso}}(\lambda) + f_{\text{vol}}(\lambda) K_{\text{vol}}(\theta_s, \theta_v, \varphi) + f_{\text{geo}}(\lambda) K_{\text{geo}}(\theta_s, \theta_v, \varphi) \tag{2}$$

where $\lambda$ is the wavelength, $\theta_s$ and $\theta_v$ are the solar and viewing zenith angles respectively, and $\varphi$ is the relative azimuth angle. In Eq. (2), $K_{\text{vol}}$ and $K_{\text{geo}}$ are the "Ross Thick" and "Li Sparse" kernels respectively. These are functions of the viewing and solar geometry describing the angular dependence of the surface reflectance and are controlled by the Leaf Area Index (LAI) of the underlying vegetation and by the surface roughness, respectively (Roujean et al., 1992; Wanner et al., 1995). The wavelength-dependent coefficients $f_{\text{vol}}$ and $f_{\text{geo}}$ determine the relative weight of the two kernels, whereas the coefficient $f_{\text{iso}}$ is an additional term describing isotropic scattering, i.e. scattering with a directionally constant amplitude. This term is equivalent to the classically defined surface albedo.

In the Maignan model, the polarized reflectance $R_p$ of a surface is expressed by the equation

$$R_p(\theta_s, \theta_v, \varphi) = \frac{C e^{-\nu} e^{-\tan(\Theta/2)} F_p(\frac{\Theta}{2}, n)}{4(\cos\theta_s + \cos\theta_v)} \tag{3}$$

where $\Theta$ is the scattering angle, $F_p$ is a function of the scattering angle and of the refractive index $n$ (Maignan et al., 2009), and $C$ and $\nu$ are two fitting parameters.





The neural network retrieves 20 quantities, 12 describing aerosols and 8 describing surface properties. The retrieved aerosol parameters are effective radius, effective variance, complex refractive index (assumed wavelength-independent) and aerosol optical thickness (AOT) at 550 nm for fine and coarse mode, fraction of spherical particles (FSP) for the coarse mode and aerosol layer height, defined as the peak height of an assumed Gaussian aerosol profile. The fine mode FSP and the full width

at half maximum (FWHM) of the aerosol profile are not retrieved, but are kept fixed at 100% and 2 km respectively. The retrieved surface parameters are the $f_{\mathrm{iso}}$ coefficient of Eq. (2) at each of the 5 input wavelengths, the "Ross Thick" and "Li Sparse" coefficients $f_{\mathrm{vol}}$ and $f_{\mathrm{geo}}$ (assumed wavelength-independent), and the $C$ parameter of the Maignan et al. (2009) BPDF model described in Eq. (3), whereas the $\nu$ parameter of the Maignan model has been set to 0.1. The input quantities of the neural network are the reflectance and the degree of linear polarization (DoLP) measured at 33 angles (from $0°$ to $40°$ with a step of

$2.5°$ forward and aftward) and at the 5 wavelengths mentioned above, plus the solar zenith angle, the relative azimuth angle $\varphi$ described above and the surface pressure. The radiance and DoLP measurements have been compressed using a principal component analysis (PCA) as explained in Di Noia et al. (2015). 23 principal components have been retained for radiance and 33 for DoLP.

   Approximately one million data have been used to train the neural network. The data have been created by generating

random combinations of the 20 retrieved quantities, solar zenith angle, relative azimuth angle and surface pressure and using such combinations as inputs for radiative transfer simulations, using a radiative transfer model described in Hasekamp and Landgraf (2002, 2005). During the training process, the synthetic measurements have been perturbed with additive Gaussian noise with zero mean and standard deviation of 2% for reflectance and 0.2% for DoLP, respectively. The angular correlation of noise in RSP measurements (Knobelspiesse et al., 2012) has not been taken into account.

In order to generate as realistic as possible statistical distributions for most of the quantities to be retrieved, we have generated global datasets of fine and coarse mode AOT and effective radius by collecting data from all the AERONET stations located over land, and we have randomly sampled the MODIS MCD43C3 product (Schaaf and Wang, 2015) in order to generate a global dataset for the $f_{\mathrm{iso}}$ coefficient in Eq. (2). For all the other parameters we have assumed a uniform statistical distribution. The main features (maximum, minimum, mean and standard deviation and type of distribution) of the statistical distributions

of all the aerosol and surface parameters used to generate the training set are summarized in Table 1. The distributions of fine and coarse mode AOT and effective radius and those of the $f_{\mathrm{iso}}$ coefficient at RSP wavelengths are indicated in the table with the term "empirical". Histograms of such empirical distributions are shown in Figures 1 and 2 for aerosol and surface parameters, respectively. It must be noted that the $f_{\mathrm{iso}}$ coefficients at the five RSP wavelengths are not statistically independent. Their correlation matrix is also plotted in Figure 2. The choice of keeping the spectral correlation of $f_{\mathrm{iso}}$ during the sampling of

the MODIS dataset was made consciously, with the aim of introducing a constraint in the neural network retrieval. In addition to the quantities listed in Table 1, also solar zenith angle, relative azimuth angle and surface pressure for the radiative transfer simulations have been chosen randomly, by assuming uniform distributions in the intervals $20°$-$85°$, $0°$-$60°$ and 850-1050 hPa respectively. A constant flight altitude of 19 km has been assumed in the simulations, as the actual flight altitude of the ER-2 aircraft hosting RSP is usually not far from this value.



**Table 1.** Details of the statistical distributions of the aerosol and surface parameters used to generate the training dataset. Please refer to Figures 1 and 2 for the histograms of the distributions indicated as "empirical".

| Parameter | Min | Max | Mean | Std dev. | Distribution |
|---|---|---|---|---|---|
| Eff. radius ($\mu$m) – fine | 0.05 | 0.46 | 0.15 | 0.04 | Empirical |
| Eff. variance – fine | 0.1 | 0.3 | 0.2 | 0.06 | Uniform |
| Refr. index (real) – fine | 1.3 | 1.7 | 1.49 | 0.10 | Uniform |
| Log. refr. index (im.) – fine | −11.5 | −0.5 | −5.75 | 3.03 | Uniform |
| AOT (550 nm) – fine | 0.0 | 4.58 | 0.13 | 0.18 | Empirical |
| Eff. radius ($\mu$m) – coarse | 0.92 | 6.12 | 2.16 | 0.45 | Empirical |
| Eff. variance – coarse | 0.4 | 0.6 | 0.5 | 0.06 | Uniform |
| Refr. index (real) – coarse | 1.3 | 1.7 | 1.49 | 0.1 | Uniform |
| Log. refr. index (im.) – coarse | −11.5 | −0.5 | −5.75 | 3.03 | Uniform |
| AOT (550 nm) – coarse | 0.0 | 3.95 | 0.06 | 0.11 | Empirical |
| Spherical fraction – coarse | 0.0 | 1.0 | 0.5 | 0.29 | Uniform |
| Aerosol layer height (m) | 250.0 | 8000.0 | 4125.0 | 2237.84 | Uniform |
| Li-Sparse BRDF parameter | 0.0 | 0.25 | 0.12 | 0.07 | Uniform |
| Ross-Thick BRDF parameter | 0.0 | 1.5 | 0.75 | 0.43 | Uniform |
| Maignan BPDF parameter | 0.02 | 10.0 | 4.83 | 2.86 | Uniform |
| Isotropic scatt. coeff. (410 nm) | 0.0 | 0.89 | 0.04 | 0.04 | Empirical |
| Isotropic scatt. coeff. (470 nm) | 0.0 | 0.90 | 0.07 | 0.05 | Empirical |
| Isotropic scatt. coeff. (550 nm) | 0.0 | 0.90 | 0.11 | 0.07 | Empirical |
| Isotropic scatt. coeff. (670 nm) | 0.0 | 0.87 | 0.16 | 0.11 | Empirical |
| Isotropic scatt. coeff. (865 nm) | 0.0 | 0.80 | 0.28 | 0.11 | Empirical |

The adopted neural network model is a multilayer perceptron (Werbos, 1974) with three hidden layers of 40 neurons each, same as the groundSPEX network presented in Di Noia et al. (2015), and the training has been carried out using the standard backpropagation algorithm (Rumelhart et al., 1986) regularized through learning rate annealing (Bös and Amari, 1999) and noise injection during training (Bishop, 1995b).

The trained neural network has been tested on about $2 \times 10^5$ random simulated data not included in the training dataset. In Table 2 the mean error (bias), the root mean square (RMS) error, the mean absolute error (MAE) and the Pearson correlation coefficient are reported for each of the retrieved aerosol and surface parameters. The statistics for the fine mode microphysical parameters are computed on the test data for which the fine mode AOT was larger than 0.1, and the same holds for the coarse mode parameters. The aerosol layer height error statistics are computed only on the cases with total (fine + coarse mode) AOT larger than 0.1. Fine mode AOT and effective radius appear to be the most accurately retrieved aerosol parameters. Good accuracies are also observed for coarse mode AOT and for aerosol layer height. Also the surface parameters are generally well retrieved. Coarse mode effective radius and effective variance appear to be the most problematic parameters, and it can be said





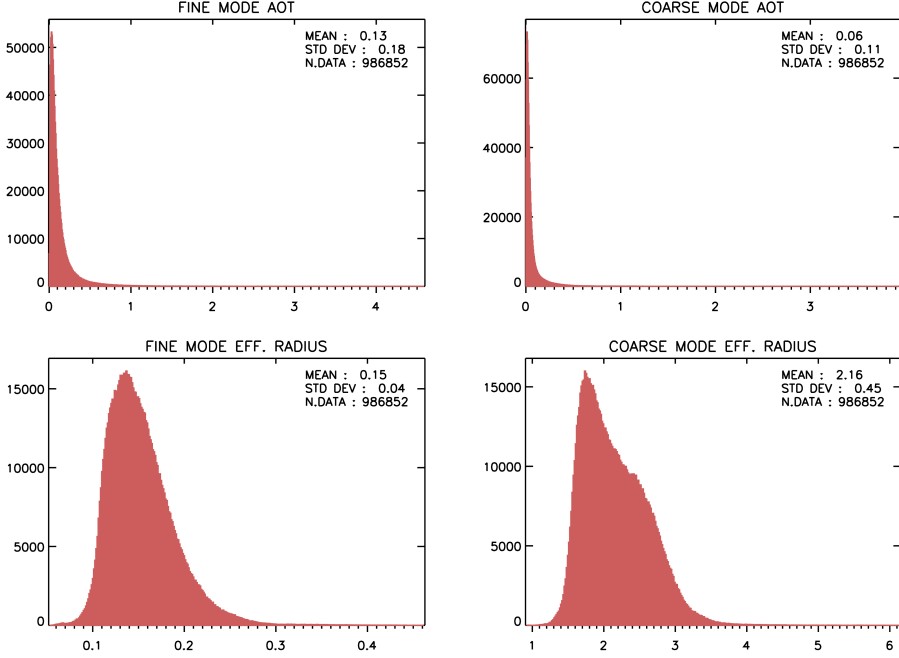

**Figure 1.** Histograms of the statistical distribution of fine and coarse mode AOT and effective radius used to generate the training dataset.

that no retrieval capability is displayed by the neural network for these two parameters. In general, the retrieval accuracy for coarse mode aerosol parameters seems worse than for fine mode parameters. This is possibly due to the particular choice of the statistical distribution of the training data, where coarse mode dominated scenarios are under-represented with respect to fine mode dominated cases. The prevalence of cases with a small total aerosol load in the training set also explains the good

5   retrieval capability of the neural network for surface BRDF and BPDF parameters.

Scatter plots of the fine and coarse mode AOT and of the fine mode effective radius are shown in Figures 3 and 4. Figure 5 shows scatter plots of an average between fine and coarse mode complex refractive index weighted by the AOTs of the two modes. The plots are computed including data with total AOT larger than 0.1. Figures 6 and 7 show scatter plots of retrieved versus true BRDF and BPDF parameters and $f_{\mathrm{iso}}$ coefficients at all wavelengths, respectively.



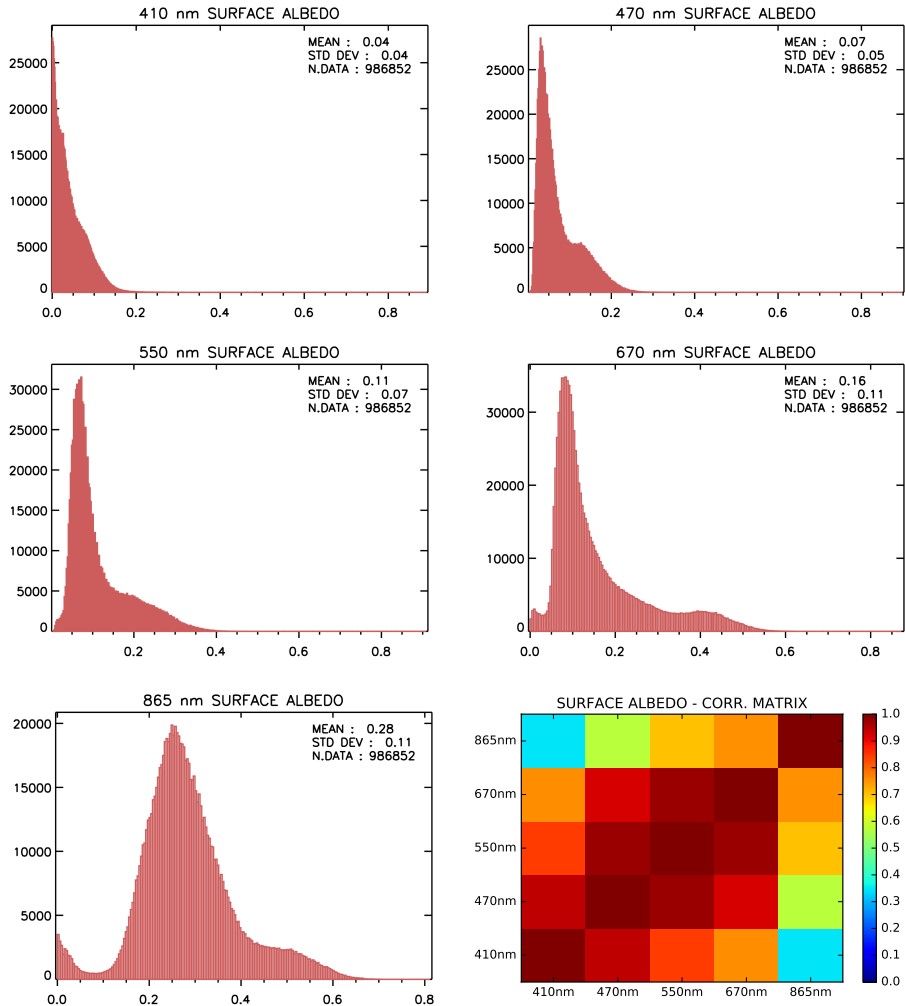

**Figure 2.** Histograms of the statistical distribution of the $f_{\mathrm{iso}}$ coefficient in the Ross-Li model at all the wavelengths used to generate the training dataset, and correlation matrix between the values of the coefficient at the used wavelengths.





**Table 2.** Error statistics on the aerosol and surface parameters retrieved by the neural network, computed on a set of data not used during the training phase.

| Parameter | Bias | RMSE | MAE | Corr. |
|---|---|---|---|---|
| Eff. radius ($\mu$m) – fine | −0.002 | 0.02 | 0.013 | 0.86 |
| Eff. variance – fine | −0.005 | 0.04 | 0.036 | 0.63 |
| Refr. index (real) – fine | 0.006 | 0.06 | 0.043 | 0.84 |
| Refr. index (im.) – fine | −0.006 | 0.05 | 0.023 | 0.89 |
| AOT (550 nm) – fine | −0.003 | 0.07 | 0.029 | 0.93 |
| Eff. radius ($\mu$m) – coarse | −0.043 | 0.45 | 0.364 | 0.13 |
| Eff. variance – coarse | −0.001 | 0.06 | 0.050 | 0.02 |
| Refr. index (real) – coarse | 0.009 | 0.08 | 0.061 | 0.71 |
| Refr. index (im.) – coarse | −0.022 | 0.09 | 0.038 | 0.63 |
| AOT (550 nm) – coarse | −0.008 | 0.06 | 0.030 | 0.82 |
| Spherical fraction – coarse | −0.016 | 0.25 | 0.207 | 0.50 |
| Aerosol layer height (m) | 42.424 | 1225.19 | 963.06 | 0.84 |
| Li-Sparse BRDF parameter | −0.002 | 0.02 | 0.015 | 0.95 |
| Ross-Thick BRDF parameter | −0.008 | 0.13 | 0.079 | 0.96 |
| Maignan BPDF parameter | −0.086 | 0.79 | 0.46 | 0.96 |
| Isotropic scatt. coeff. (410 nm) | −0.001 | 0.01 | 0.006 | 0.96 |
| Isotropic scatt. coeff. (470 nm) | −0.001 | 0.01 | 0.006 | 0.98 |
| Isotropic scatt. coeff.(550 nm) | −0.001 | 0.02 | 0.009 | 0.98 |
| Isotropic scatt. coeff. (670 nm) | −0.001 | 0.02 | 0.010 | 0.99 |
| Isotropic scatt. coeff.(865 nm) | −0.001 | 0.02 | 0.014 | 0.98 |





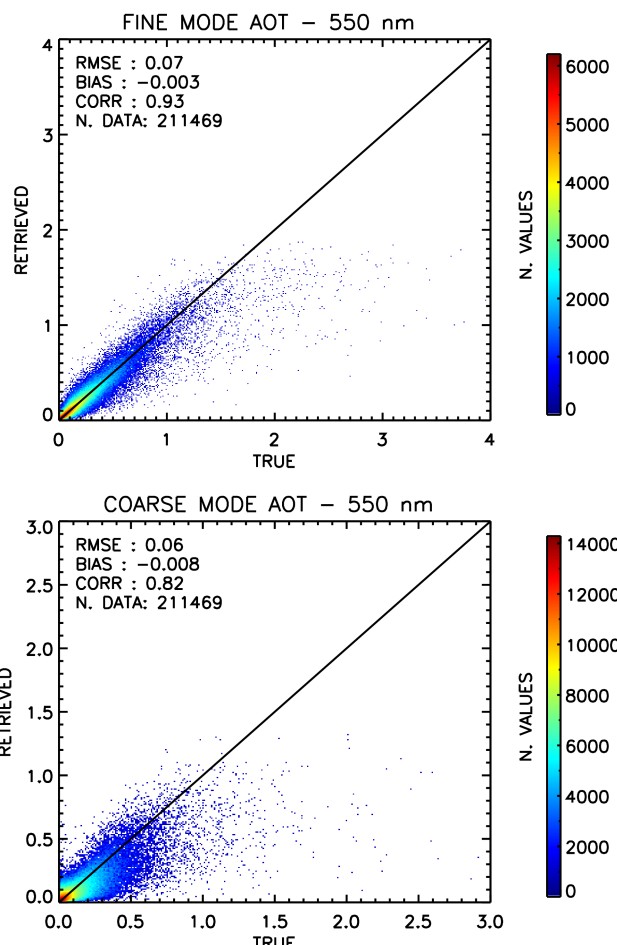

**Figure 3.** Neural network retrieved versus true fine and coarse mode AOT from synthetic test data.



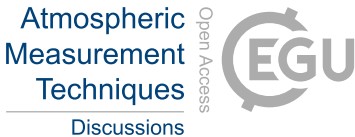

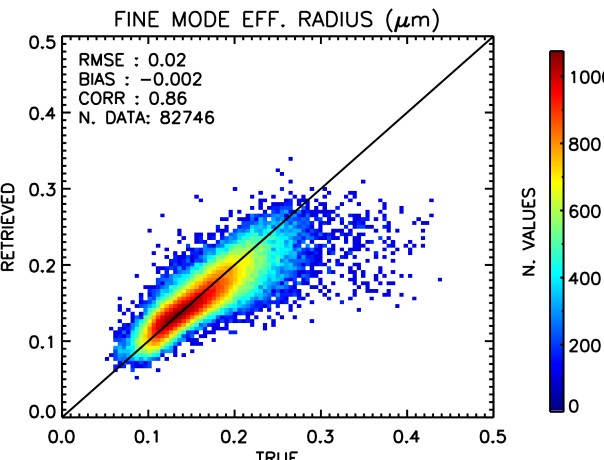

**Figure 4.** Neural network retrieved versus true fine mode effective radius from synthetic test data. Only cases with fine mode AOT larger than 0.1.





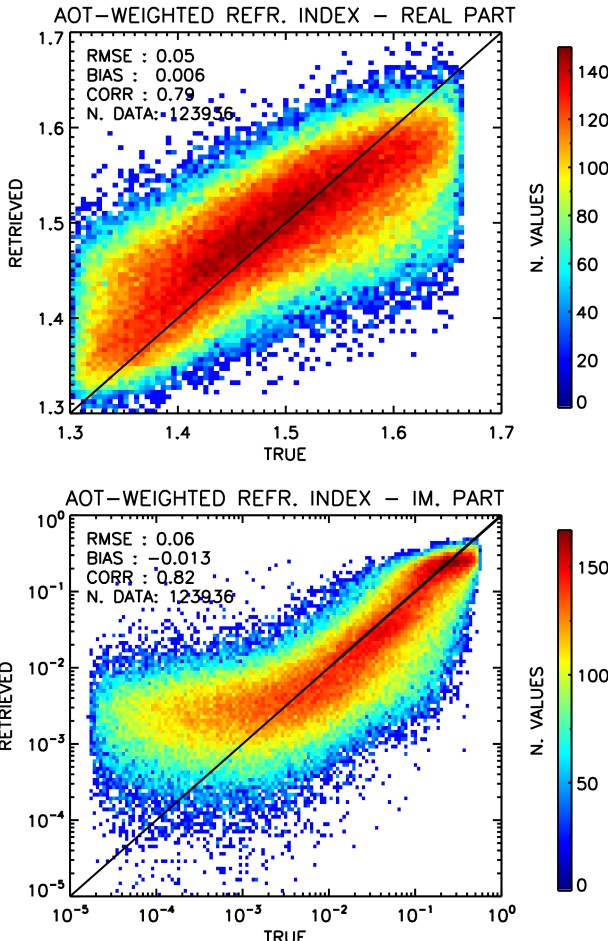

**Figure 5.** Neural network retrieved versus true real and imaginary part of the refractive index, weighted by mode AOT. Synthetic data. Only cases with total AOT larger than 0.1



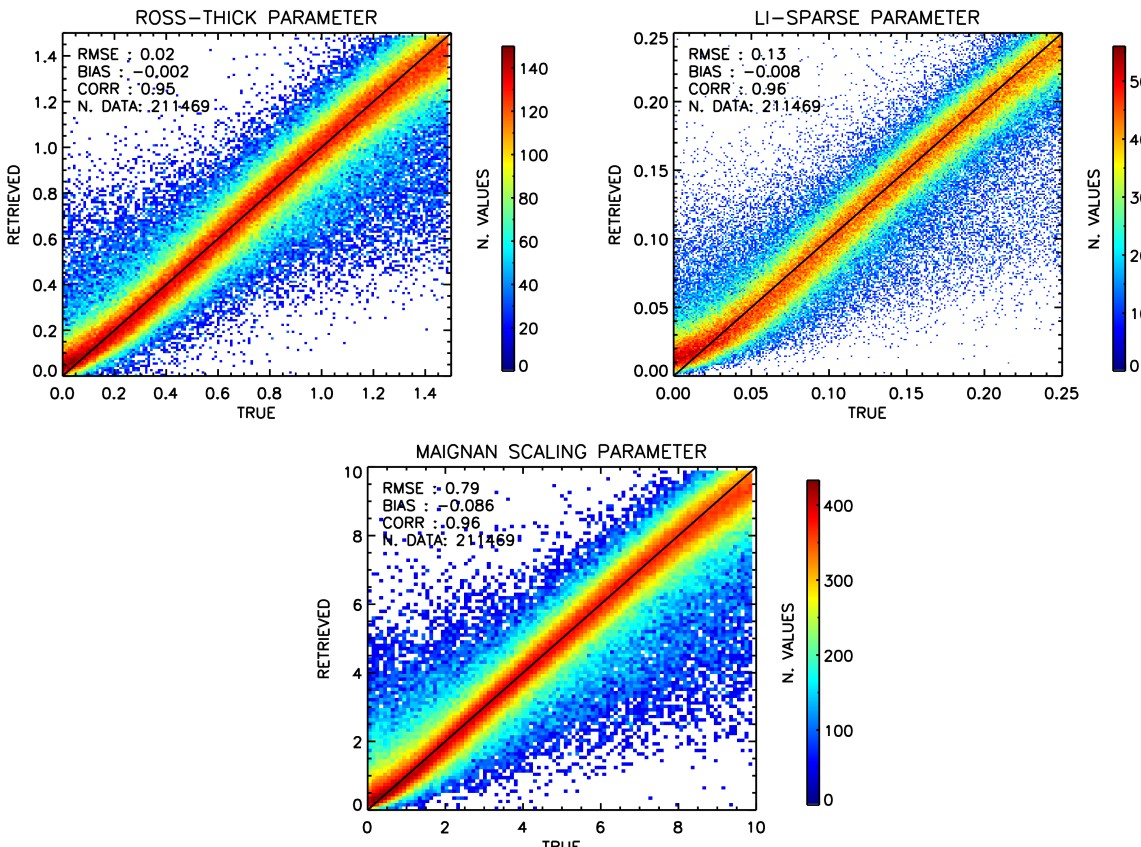

**Figure 6.** Neural network retrieved versus true Ross-Thick and Li-Sparse BRDF parameters and Maignan BPDF parameter from synthetic test data.




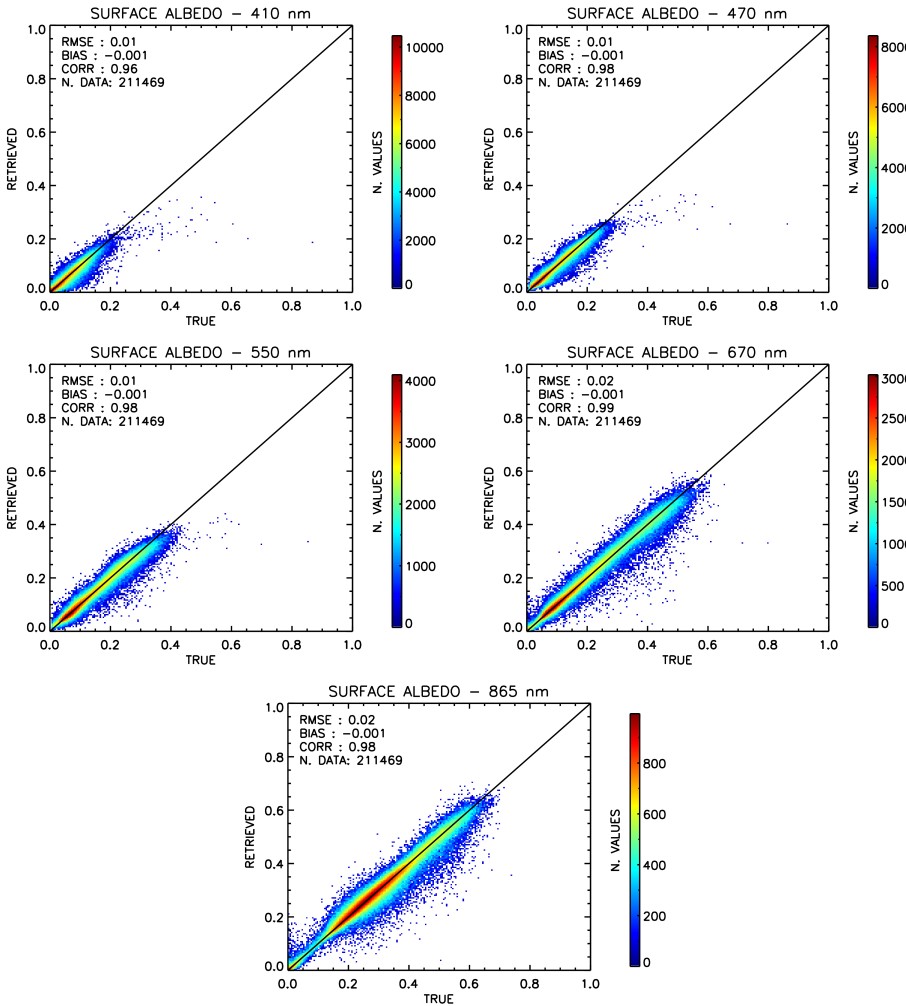

**Figure 7.** Neural network retrieved versus true $f_{\mathrm{iso}}$ coefficients from synthetic test data, at the five wavelengths considered in this study.





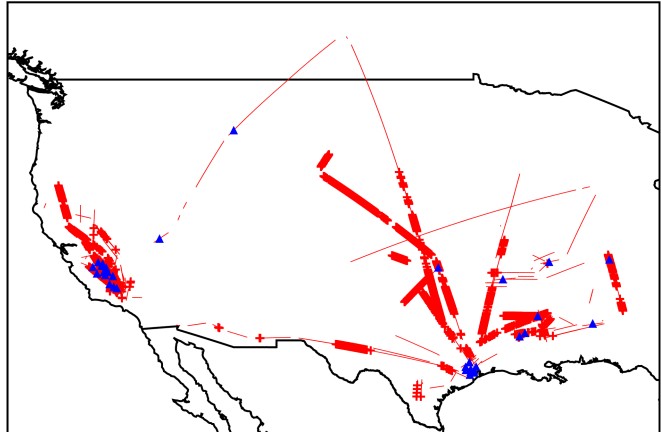

**Figure 8.** Map of the RSP data used in this study. The red thin lines indicate RSP flights, the red crosses indicate the availability of CPL data and the blue triangles indicate the locations of the AERONET stations.

## 5 Application to RSP measurements

The neural network scheme described in Section 4 has been used as initial guess for a Phillips-Tikhonov retrieval scheme described in Wu et al. (2015) in order to process a collection of RSP measurements carried out during the PODEX and SEAC[4]RS campaigns which took place in 2012 and 2013 over several areas of the United States and part of Canada. A total of

5     7770 measurements have been collected, and the data of 36 AERONET stations have been used for validation. Furthermore, data from the Cloud and Physics Lidar (CPL, McGill et al., 2002), hosted on the NASA ER-2 aircraft together with RSP, have been used to validate aerosol layer height retrievals and to perform cloud screening. A map of the considered flights and of the AERONET stations used for validation is shown in Fig. 8 and the full list of AERONET stations providing validation data is given in Table 3.

10     In order to reduce the angular oscillations in RSP measurements due to imperfect coregistration, two pre-processing steps have been applied to RSP measurements before the application of the retrieval algorithm. First, the measurements have been averaged over a 5 km box, as in Wu et al. (2015). Second, each horizontally averaged multiangular RSP measurement has been convolved with a moving-average Blackman filter of length 5. As explained in Section 3, only the measurements made at VZAs between $40°$ aftward and $40°$ forward have been used as input for the neural network. The measurement vector for

15     the Phillips-Tikhonov retrieval, instead, covers the entire VZA range of RSP. The Phillips-Tikhonov scheme uses 9 viewing angles, selected on a measurement-by-measurement basis in order to cover a scattering angle range that is as broad as possible,



**Table 3.** List of the AERONET stations used for validation in this study.

| Station name | Lat. | Lon. |
|---|---|---|
| Baskin | 32.28 | -91.74 |
| Bozeman | 45.66 | -111.04 |
| Caldwell Parish HS | 32.06 | -92.10 |
| CART Site | 36.61 | -97.49 |
| DRAGON Aldine | 29.90 | -95.33 |
| DRAGON Arvin | 35.24 | -118.79 |
| DRAGON Bakersfield | 35.33 | -119.00 |
| DRAGON Channel View | 29.80 | -95.13 |
| DRAGON Clinton | 29.73 | -95.26 |
| DRAGON Clovis | 36.82 | -119.72 |
| DRAGON Conroe | 30.35 | -95.43 |
| DRAGON Corcoran | 36.10 | -119.57 |
| DRAGON Deer Park | 29.67 | -95.13 |
| DRAGON Drummond | 36.71 | -119.74 |
| DRAGON Garland | 36.78 | -119.77 |
| DRAGON Hanford | 36.32 | -119.64 |
| DRAGON Huron | 36.21 | -120.11 |
| DRAGON Madera City | 36.95 | -120.03 |
| DRAGON ManvelCroix | 29.52 | -95.39 |
| DRAGON NW Harris Co. | 30.04 | -95.67 |
| DRAGON Parlier | 36.60 | -119.50 |
| DRAGON Porterville | 36.03 | -119.06 |
| DRAGON Shafter | 35.50 | -119.27 |
| DRAGON Tranquility | 36.63 | -120.38 |
| DRAGON UH W Liberty | 30.06 | -94.98 |
| DRAGON Visalia | 36.31 | -119.39 |
| DRAGON West Houston | 29.83 | -95.66 |
| Fresno-2 | 36.78 | -119.77 |
| IMPROVE-MammothCave | 37.13 | -86.15 |
| Leland HS | 33.40 | -90.89 |
| Mingo | 36.97 | -90.14 |
| Railroad Valley | 38.50 | -115.96 |
| SEARCH Centreville | 32.90 | -87.25 |
| SEARCH Centreville2 | 32.90 | -87.25 |
| Univ. of Houston | 29.72 | -95.34 |
| Upper Buffalo | 35.83 | -93.20 |





thereby maximizing the sensitivity of the spectropolarimetric measurements to the aerosol properties (Wu et al., 2015). The Phillips-Tikhonov retrieval scheme performs an iterative minimization of the following cost function:

$$C(\boldsymbol{x}) = [\boldsymbol{y} - \boldsymbol{F}(\boldsymbol{x})]^T \boldsymbol{S}_\epsilon^{-1} [\boldsymbol{y} - \boldsymbol{F}(\boldsymbol{x})] + \gamma(\boldsymbol{x} - \boldsymbol{x}_a)\boldsymbol{H}(\boldsymbol{x} - \boldsymbol{x}_a) \tag{4}$$

In Eq. (4), $\boldsymbol{x}$ is the state vector, $\boldsymbol{x}_a$ is an a priori state vector, $\boldsymbol{y}$ is the measurement vector, $\boldsymbol{F}(\boldsymbol{x})$ is the simulated measurement
vector corresponding to the state $\boldsymbol{x}$, $\boldsymbol{S}_\epsilon$ is the measurement error covariance matrix, $\boldsymbol{H}$ is a regularization matrix and $\gamma$ is a regularization parameter, which is determined using the L-curve method (Hansen and O'Leary, 1993). The metric used to evaluate the convergence of a Phillips-Tikhonov retrieval is the so-called goodness-of-fit parameter

$$\chi^2 = \frac{1}{m} \sum_{i=1}^{m} \frac{[y_i - F_i(\boldsymbol{x})]^2}{\sigma_i^2} \tag{5}$$

where $m$ is the dimension of the measurement vector, $y_i$ and $F_i(\boldsymbol{x})$ are the $i$-th components of $\boldsymbol{y}$ and $\boldsymbol{F}(\boldsymbol{x})$ respectively, and
$\sigma_i$ is the standard deviation of the measurement error for $y_i$ ($i = 1, \ldots, m$). We empirically found that a retrieval can be said to have converged successfully if it achieves a goodness-of-fit parameter smaller than 2.

For the purpose of validating AOT retrievals, we considered each RSP measurement as co-located with an AERONET measurement if the distance between RSP and AERONET was not larger than 5 km and the measurements were taken no longer than 1 hour apart. For validation of aerosol properties other than AOT we had to relax the co-location criterion by
allowing for a threshold of 20 km in distance and one day in time. This was necessary in order to ensure the availability of an adequate number of data points. Additional criteria imposed for validation were the following: (i) minimum scattering angle of RSP measurements not larger than $85°$; (ii) only RSP measurements flagged as cloud-free from CPL (cloud flag equal to 0) were considered; (iii) thresholds on AOT (explained later in this section) were applied when validating aerosol properties other than the AOT. Out of the original dataset consisting of 7770 measurements, 2327 measurements satisfied the screening criteria
(i) and (ii).

In total we found 95 AOT retrievals fulfilling the co-location criteria with distance threshold set to 5 km and 825 retrievals with the looser threshold of 20 km. 25 co-located retrievals satisfied the goodness-of-fit criterion mentioned earlier in this section ($\chi^2 < 2$) with 5 km threshold, whereas 103 retrievals met the same criterion with the 20 km threshold. The total number of converging retrievals (i.e., including non co-located retrievals) was 223 out of the 2327 retrievals satisfying the
criteria mentioned above on CPL cloud flag and minimum scattering angle. Thus, to summarize, approximately 10% of the retrievals achieved a $\chi^2$ smaller than 2. A histogram of the $\chi^2$ of all the retrievals is shown in Fig. 9. Approximately 75% of the retrievals reached a $\chi^2$ less than 10, which is the convergence threshold used in Wu et al. (2015). Retrievals with a larger $\chi^2$, which form the tail of the distribution, are probably due to measurements still affected by angular oscillations after the filtering described earlier in this section.
Scatter plots of the retrieved AOTs versus those provided by the AERONET Level 2 product based on the direct-sun algorithm (Holben et al., 1998) at four AERONET wavelengths are shown in Fig. 10. The results of the neural network alone as well as those of the Phillips-Tikhonov retrieval scheme are shown. The RMS errors of neural network retrievals range between





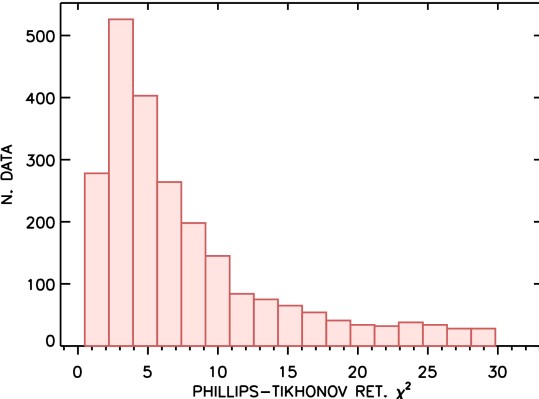

**Figure 9.** Histogram of the goodness-of-fit parameter ($\chi^2$) of all the available retrievals.

**Table 4.** RMS difference between RSP and AERONET in the AOTs retrieved using the Phillips-Tikhonov scheme initialized by neural network at four wavelengths, as a function of the maximum tolerated distance for co-location.

| Max. distance | N. data | 440 nm | 500 nm | 675 nm | 870 nm |
|---|---|---|---|---|---|
| 5 km | 25 | 0.044 | 0.044 | 0.045 | 0.045 |
| 10 km | 46 | 0.047 | 0.047 | 0.051 | 0.052 |
| 15 km | 71 | 0.067 | 0.068 | 0.070 | 0.069 |
| 20 km | 103 | 0.071 | 0.072 | 0.074 | 0.074 |

approximately 0.04 at 870 nm and 0.07 at 440 nm. The application of the Phillips-Tikhonov retrievals brings the RMS error down to about 0.045 at all wavelengths (see Table 5). Looking at the plots, it appears that the Phillips-Tikhonov retrieval brings an improvement over the neural network retrieval in cases with AOT at 440 nm around 0.3 and larger. The neural network seems to underestimate the AOT in these cases, whereas the Phillips-Tikhonov seems to bring the retrieved values closer to the AERONET values. It is worthwhile to note that the error statistics seem to depend critically on the distance threshold chosen for co-location, as shown in Table 4, from which a steady increase in the RMS error is evident as the co-location criterion is relaxed.

A comparison between RSP and AERONET 440-675 nm Ångström exponents, restricted to cases with retrieved AOT at 440 nm larger than 0.1, is shown in Fig. 11. From the figure it is evident that RSP retrieves systematically lower Ångström exponents than indicated by AERONET, which means that our retrieval algorithm may have a tendency to overestimate the size of aerosol particles.

A good agreement between RSP and AERONET is observed for the fine mode effective radius derived from the Level 2 almucantar product (Dubovik and King, 2000; Dubovik et al., 2000). The results of the comparison – which is limited to cases with retrieved AOT at 440 nm larger than 0.2 – are shown in Fig. 12. The RMS errors are slightly lower than 0.03 $\mu$m for both



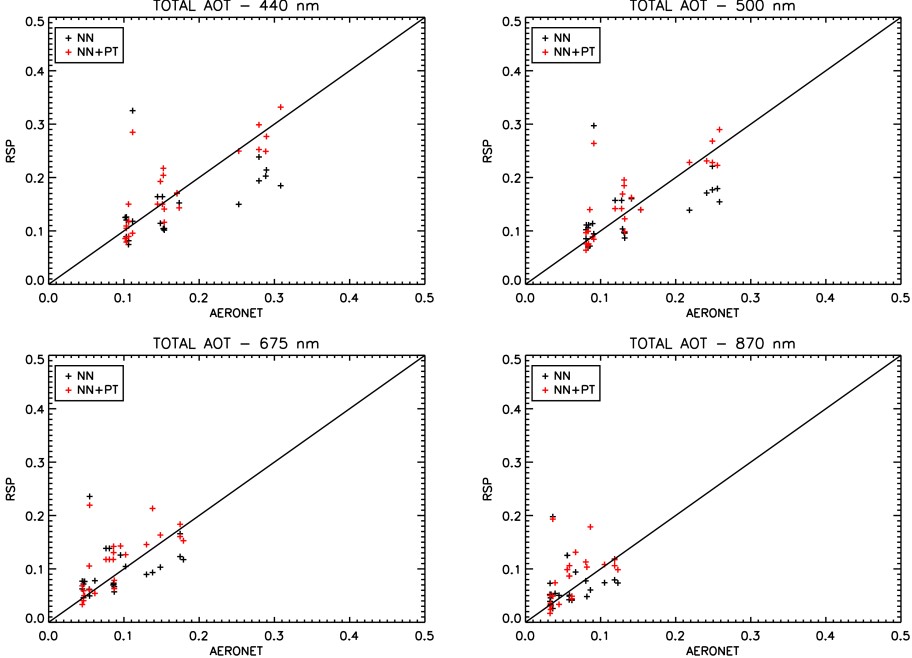

**Figure 10.** RSP versus AERONET total AOT at the four AERONET wavelengths lying inside the spectral range of the RSP measurements used in this study. Black crosses: neural network retrievals. Red crosses: Phillips-Tikhonov with neural network first guess. Statistics shown in Table 5.

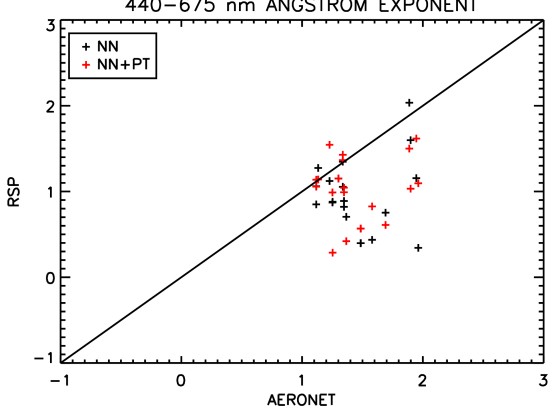

**Figure 11.** RSP versus AERONET 440-675 nm Ångström exponent. Black crosses: neural network retrievals. Red crosses: Phillips-Tikhonov with neural network first guess. Statistics shown in Table 5.





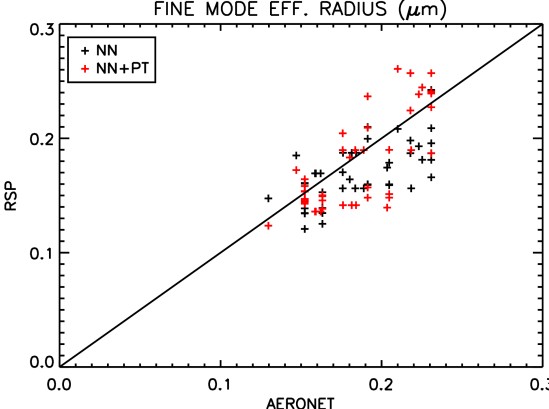

**Figure 12.** RSP versus AERONET fine mode effective radius. Black crosses: neural network retrievals. Red crosses: Phillips-Tikhonov with neural network first guess. Statistics shown in Table 5.

the neural network retrieval and the final retrieval. As mentioned before, the presence of an additional filter on AOT required us to relax the co-location distance to 20 km in order to obtain a reasonable amount of data points. For this co-location criterion we obtained 46 data points, whereas adopting a stricter criterion of 10 km the number of data would drastically drop to 16. Comparisons for complex refractive index and single scattering albedo (SSA) are not shown because no AERONET data points

satisfying the quality assurance criteria for Level 2 products (Holben et al., 2006) were found.

A comparison between the retrieved aerosol layer heights and CPL measurements has been made by taking into account all the cases in which the retrieved AOT at 440 nm was larger than 0.1. In Fig. 13 a plot of retrieved aerosol heights versus coincident CPL measurements is shown, with 141 coincident retrievals that passed the convergence and the other screening criteria. It is possible to see that the neural network retrievals alone are already in good agreement with the CPL, with a RMS

error around 1670 m and a correlation coefficient of 0.76. No clear improvement is observed over the neural network retrieval after the Phillips-Tikhonov algorithm is applied. On the contrary, a slight degradation of the validation statistics is observed (RMS difference around 1735 m and correlation coefficient around 0.75), but it is doubtful that these differences bear any statistical significance. The results of the comparison seem also consistent with those shown in Wu et al. (2016), where an iterative algorithm with multiple LUT-based initialization was adopted.

The full statistics regarding bias, RMS error and correlation coefficient for the comparisons shown in Figs. 10 to 13 are shown in Table 5.

Compared to the traditional algorithm initialized by a LUT (Hasekamp et al., 2011; Wu et al., 2015), the algorithm initialized by means of the neural network is characterized by a larger number of converging retrievals and a lower processing time. By applying the LUT-based algorithm to the dataset using in this study we obtained in total 47 converging retrievals (223 with

the neural network). The time required in order to process the entire dataset can be reduced by roughly a factor 2 when using the neural network algorithm to provide the first guess. The differences in the retrieved aerosol parameters between the LUT-





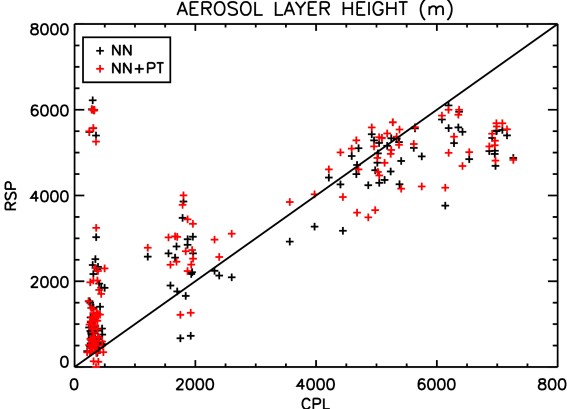

**Figure 13.** RSP versus CPL aerosol layer height. Black crosses: neural network retrievals. Red crosses: Phillips-Tikhonov with neural network first guess. Statistics shown in Table 5.

**Table 5.** Error statistics from the comparisons shown in Figs. 10 to 13. for the neural network(NN) and for the Phillips-Tikhonov retrieval initialized using the neural network (NN+PT).

| Parameter | N.data | NN | | | NN+PT | | |
| --- | --- | --- | --- | --- | --- | --- | --- |
| | | Bias | RMSE | Corr. | Bias | RMSE | Corr. |
| AOT (440 nm) | 25 | −0.02 | 0.07 | 0.51 | 0.01 | 0.04 | 0.82 |
| AOT (500 nm) | 25 | −0.01 | 0.06 | 0.48 | 0.02 | 0.04 | 0.80 |
| AOT (675 nm) | 25 | 0.01 | 0.05 | 0.35 | 0.02 | 0.04 | 0.69 |
| AOT (870 nm) | 25 | 0.01 | 0.04 | 0.18 | 0.02 | 0.04 | 0.50 |
| Ångström exp. (440-675 nm) | 20 | −0.50 | 0.68 | 0.11 | −0.44 | 0.61 | 0.14 |
| Fine mode eff. radius [$\mu$m] | 46 | −0.02 | 0.03 | 0.66 | −0.01 | 0.03 | 0.76 |
| Aerosol layer height [km] | 141 | 0.45 | 1.67 | 0.76 | 0.58 | 1.73 | 0.75 |

based algorithm and the NN-based do not seem significant. The same tendency to overestimate coarse mode aerosol loads is also observed in LUT-based retrievals. Plots of retrieved versus AERONET (or CPL) aerosol parameters for the LUT-based algorithm are given in the supplementary material.

## 6 Conclusions

In this paper we have demonstrated the application of neural networks to aerosol retrievals from the RSP instrument. First, we trained a neural network to retrieve aerosol and surface properties from simulated multiangle spectropolarimetric measurements, with an observation geometry that simulates that of RSP. Then we have applied the neural network to real RSP





measurements carried out over the United States in 2012 and 2013, and we have used the neural network retrieval as first guess for an iterative algorithm based on the Phillips-Tikhonov method.

In order to assess the quality of our retrievals, we compared the retrieved aerosol parameters and aerosol layer heights to co-located AERONET and CPL lidar measurements, respectively. We observed good retrieval capability for AOT (although

with a possible overestimation of the coarse mode), fine mode effective radius and aerosol layer height. A comparison of the retrieval results with those obtained using an iterative algorithm initialized using a LUT confirms the finding that replacing the LUT with a neural network generally leads to a higher number of converging retrievals and to a reduced processing time.

Training a neural network algorithm for multi-parameter aerosol retrievals from airborne or satellite multi-angle spectropolarimetric measurements remains challenging. In our case, for example, the use of a training set encompassing mostly fine

mode dominated cases with moderate to low aerosol loads resulted in better retrieval capabilities for the fine mode aerosol parameters than for the coarse mode parameters, and in good retrieval capabilities for all the surface parameters. Choosing a mixture of training cases for a single neural network that allows to achieve equal performances for all the aerosol and surface parameters on fine mode as well as on coarse mode dominated situations, or with low as well as with high aerosol loads, may be a challenging task. A possible improvement over the present work may involve training multiple neural networks, each cover-

ing a certain range of situations, with possibly an additional neural network that performs a preliminary classification, thereby deciding which of the lower level networks should be used for retrieval. Approaches of this kind have been already applied to simple inverse scattering problems involving spheres and spheroids (Ulanowski et al., 1998; Berdnik et al., 2004), and may be as well investigated – with due adaptations – for more complex retrievals. The aforementioned approach may be also useful for extending the method presented in this paper to multi-angle spectropolarimeters in which the angular dependence of the

measurements is more complicated, such as POLDER, AirMSPi and the forthcoming 3MI, where it is not possible to assume that all the measurements are made at the same relative azimuth angle apart from shifts of $180°$.

*Acknowledgements.* The RSP data used in this study are publicly available from the NASA Goddard Institute for Space Studies (http://data. giss.nasa.gov/pub/rsp/). The RSP data from the SEAC[4]RS and PODEX field experiments were funded by the NASA Radiation Sciences Program managed by Hal Maring and by the NASA Earth Science Division, as part of the pre-formulation study for the Aerosol Cloud and

ocean Ecosystem (ACE) mission.

The CPL data are provided by the NASA Goddard Space Flight Center (http://cpl.gsfc.nasa.gov/).

We thank the Principal Investigators (PIs) and their staff for establishing and maintaining the AERONET sites whose data have been used in this paper.

The MODIS Terra+Aqua MCD43C3 BRDF/Albedo L3 data product was retrieved from the online Data Pool, courtesy of the NASA Land

Processes Distributed Active Archive Center (LP DAAC), USGS/Earth Resources Observation and Science (EROS) Center, Sioux Falls, South Dakota, United States, https://lpdaac.usgs.gov/data_access/data_pool.





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
