# Peer review of "Combined neural network / Phillips-Tikhonov approach to aerosol retrievals over land from the NASA Research Scanning Polarimeter"

_Atmospheric Measurement Techniques, 2017_

## Referee Comment (RC1) · Anonymous Referee #2 · 31 Jul 2017

The manuscript should be very interesting for broad scientific community working with satellite data retrieval. The application of neural networks as an approach for aerosol/surface retrieval looks very promising. The results presented in the manuscript are convincing and well-described. The manuscript is completely suitable for publication in AMT.

Application of the neural network approach to airborne RSP measurements shows performance comparable with Phillips-Tikhonov approach with neural network first guess. This result seems a little bit confusing and requires more discussions in the manuscript, taking into account big potential of Phillips-Tikhonov approach with neural network first

guess. It would be useful if authors could provide also some results of the synthetic test data retrieval both with Phillips-Tikhonov approach and the neural network approach. In this case it would be possible to compare two approaches on the same controlled synthetic data set.

Technical remarks: 1. Figures 10-13 do not contain any statistical characteristics like RMSE, BIAS, Correlation coefficient, number of used data. Adding these characteristics similarly to Figures 3-7 will make the presented results more clear. 2. On page 18 it is written: "We empirically found that a retrieval can be said to have converged successfully if it achieves a goodness-of-fit parameter smaller than 2." Looking at Table 4 and Figure 9, one can conclude that the percentage of the "converged retrieval pixels" is very small. To better understand the convergence of the approach, the percentage of "converged retrieval pixels" would be very useful as additional parameter, for example, in Table 4.

Overall, I recommend this paper for publication in AMT after discussion and review corrections.

---

## Referee Comment (RC2) · A. Lyapustin (Referee) · 4 Aug 2017

This is a well-designed and well-written study. The neural network (NN) is trained based on the radiative transfer simulations first, and then used to arrive at first guess solution for the following Phillips-Tikhonov minimization when processing RSP data. The NN-accuracy is demonstrated based on synthetic data, and the algorithm is applied to process PODEX and SEAC4RS flight campaign data. The paper is a good contribution to the field, and should be published after authors make a couple of corrections below. I have just one question which should be outlined, perhaps, in the Abstract or summary, and was not really clear to me after reading the paper. Of all

field campaign data, what % of experiments did you process in the end? Paper says ∼10% based on convergence to chi2<2. From chi2>2, what % is due to failure from the surface retrievals? You can evaluate chi2 from the surface alone based on simulated experiments. My feeling is that adding surface spectral covariance as a constraint may not serve you well. Also, the retrieval accuracy of ∼0.01 surface reflectance (perhaps larger since 0.01 is rmse) in the visible bands is not good enough for the land applications, e.g. vegetation studies, and it creates a considerable uncertainty for the aerosol retrieval, although of course, aerosol-surface parts are not separated in the described algorithm.

1. P.5, Ln. 12: The backscattering azimuth is 180-phi (you have 180+phi). 2. P.5, Ln.27: "This term is equivalent to the classically defined surface albedo." This is incorrect – please remove here and correct everywhere in the paper. Surface albedo is "classically" defined as a ratio of reflected and incident surface fluxes. This ratio will equal f_iso ONLY if hemispheric integrals of terms containing K_vol and K_geo in the boundary condition of RT are zero, and they are not. For the same reason, surface albedo is a function of SZA (e.g., see Lyapustin, 1999, JGR).

Sincerely, Alexei.

---

## Author Comment (AC1) · 18 Sep 2017

We thank Reviewer 2 for his/her comments. Below is our reply. The Reviewer's comments are highlighted in bold, our replies are in plain text.

**Application of the neural network approach to airborne RSP measurements shows performance comparable with Phillips-Tikhonov approach with neural network first guess. This result seems a little bit confusing and requires more discussions in the manuscript, taking into account big potential of Phillips-Tikhonov approach with neural network first guess. It would be useful if authors could provide also some results of the synthetic test data retrieval both**

**with Phillips-Tikhonov approach and the neural network approach. In this case it would be possible to compare two approaches on the same controlled synthetic data set.**

We have followed the Reviewer's suggestion. We have added a section to the revised paper, in which the results of a comparison between the combined algorithm (neural network + Phillips-Tikhonov) and the neural network alone on synthetic test data are shown.

**Figures 10-13 do not contain any statistical characteristics like RMSE, BIAS, Correlation coefficient, number of used data. Adding these characteristics similarly to Figures 3-7 will make the presented results more clear.**

We would prefer to keep the error statistics in a separate table, as it is in the current version of the manuscript. We feel that showing the statistics directly in the figures would make the figures overly filled with text. The figure captions point to the table in which the statistics are summarized, thus we believe there should be no risk of confusion for the reader.

**On page 18 it is written: "We empirically found that a retrieval can be said to have converged successfully if it achieves a goodness-of-fit parameter smaller than 2." Looking at Table 4 and Figure 9, one can conclude that the percentage of the "converged retrieval pixels" is very small. To better understand the convergence of the approach, the percentage of "converged retrieval pixels" would be very useful as additional parameter, for example, in Table 4.**

Table 4 would probably not be the best place where to summarize this percentage, because its goal is rather to show the dependence of the validation statistics on the co-location distance. At page 21 in the revised manuscript we mention that approximately 10% of the retrievals converged to a chi2 less than 2, and at page 25 we mention that this is probably not due to the neural network approach, since with the previous version of the algorithm (not employing the NN) the fraction of converging retrievals was even

smaller. In the revised version of the manuscript, we also summarize this information in the conclusions.

---

## Author Comment (AC2) · 18 Sep 2017

We are thankful to Dr. Alexei Lyapustin for his review. Below is our reply. The Reviewer's comments are highlighted in bold, our replies are in plain text.

**I have just one question which should be outlined, perhaps, in the Abstract or summary, and was not really clear to me after reading the paper. Of all field campaign data, what % of experiments did you process in the end? Paper says ~10% based on convergence to chi2<2. From chi2>2, what % is due to failure from the surface retrievals? You can evaluate chi2 from the surface alone based on simulated experiments. My feeling is that adding surface spectral covariance**

**as a constraint may not serve you well. Also, the retrieval accuracy of ∼0.01 surface reflectance (perhaps larger since 0.01 is rmse) in the visible bands is not good enough for the land applications, e.g. vegetation studies, and it creates a considerable uncertainty for the aerosol retrieval, although of course, aerosol-surface parts are not separated in the described algorithm.**

After we applied our data filtering (based on scattering angle and cloud screening) we processed 2327 RSP measurements, and about 10% of these retrievals converged to a chi2 less than 2, as you correctly mention. In the revised version of the paper we summarized this in the conclusions. We feel that the abstract is not the best part of the manuscript in which to include this information, as this is not part of the main message of the paper. At the moment we are unable to offer an quantification of the percentage of retrievals which fail to converge because of failures in the retrieval of the surface properties. Our simulated retrievals did not display major issues related to this point. We agree, however, that if, for instance, the BRDF models we use in our retrieval scheme fail to reproduce the angular behaviour of some real surfaces underlying RSP, this has the potential of leading to high chi2. Possibly, in order to evaluate how significant this effect is, we should generate synthetic data assuming surface BRDFs that deviate significantly from the Ross-Li model and try to perform the retrievals assuming the model is still valid. In our current simulation setup, though, this is not straightforward.

Regarding the accuracy of the retrieval of surface reflectance, in the revised version of the paper we added a sentence in which it is mentioned that an accuracy of 0.01 in surface reflectance is not sufficient for land applications, but may be still adequate for climate models (Wang et al., 2004, He et al., 2014, and references therein).

**P.5, Ln. 12: The backscattering azimuth is 180-phi (you have 180+phi).**

Our impression is that this is not the case. In this sentence we explain the angular relationship between an RSP measurement made in the forward direction and one

made in the aftward direction. Our modeling assumption is basically that all the RSP measurements lie on the same line. Figure 1, placed at the end of this document, gives an idea of the viewing geometry we assume in our model. The red dots represent angular measurements made in the forward direction at a given relative azimuth angle $\varphi$, whereas the blue dots represent measurements made in the aftward direction. It seems to us that for such measurements the relative azimuth angle is $\varphi + 180°$.

**P.5, Ln.27: "This term is equivalent to the classically defined surface albedo." This is incorrect – please remove here and correct everywhere in the paper. Surface albedo is "classically" defined as a ratio of reflected and incident surface fluxes. This ratio will equal $f_{iso}$ ONLY if hemispheric integrals of terms containing $K_{vol}$ and $K_{geo}$ in the boundary condition of RT are zero, and they are not. For the same reason, surface albedo is a function of SZA (e.g., see Lyapustin, 1999, JGR).**

Thank you for making us aware of this. We have removed this statement from the paper, and we have changed the titles of the figures in which the term "surface albedo" appears. We have replaced "surface albedo" with "isotropic scattering coefficient".

REFERENCES

Wang, K., Liu, J., Zhou, X., Sparrow, M., Ma, M., Sun, Z., and Jiang, W. (2004), "Validation of the MODIS global land surface albedo product using ground measurements in a semidesert region on the Tibetan Plateau", J. Geophys. Res., 109, D05107, doi: 10.1029/2003JD004229

He, T., Liang, S., and Song, D.-X. (2014), "Analysis of global land surface albedo climatology and spatial-temporal variation during 19812010 from multiple satellite products", J. Geophys. Res., 119, 10281-10298, doi: 10.1002/2014JD021667

[Figure]

[Figure]

**Fig. 1.** Idealized viewing geometry assumed to model the angular dependence of RSP measurements. Each dot in the polar plot represents a single angular measurement.